# Detecting Multilevel Manipulation from Limit Order Book via Cascaded Contrastive Representation Learning

## Abstract

Trade-based manipulation (TBM) undermines the fairness and stability of financial markets drastically. Spoofing, one of the most covert and deceptive TBM strategies, exhibits complex anomaly patterns across multilevel prices, while often being simplified as a single-level manipulation. These patterns are usually concealed within the rich, hierarchical information of the Limit Order Book (LOB), which is challenging to leverage due to high dimensionality and noise. To address this, we propose a representation learning framework combining a cascaded LOB representation architecture with supervised contrastive learning. Extensive experiments demonstrate that our framework consistently improves detection performance across diverse models, with Transformer-based architectures achieving state-of-the-art results. In addition, we conduct systematic analyses and ablation studies to investigate multilevel manipulation and the contributions of key components for detection, offering broader insights into representation learning and anomaly detection for complex time series data.

## 1 Introduction

As the backbone of modern economies, financial markets rely heavily on efficiency and integrity to ensure stable and fair operations worldwide (Roodposhti et al., 2011). However, market manipulation, particularly trade-based manipulation (TBM) as classified in (Allen & Gale, 1992), can severely undermine market fairness and erode investor confidence. Increasingly sophisticated TBM strategies have recently emerged amid the rapid growth of electronic markets and algorithmic trading. These developments pose significant challenges for regulators and have heightened concerns among market participants (Alexander & Cumming, 2022). Regulatory bodies, including the China Securities Regulatory Commission (CSRC) and the U.S. Securities and Exchange Commission (SEC), actively monitor and penalize such behaviors to uphold market integrity and safeguard investors.

One of TBM's most covert and difficult forms is spoofing (or named layering), a deceptive trading strategy involving non-*bona fide* order placements. Spoofing typically involves placing large orders without the intention of execution, often hidden in deeper levels (after the 2nd level) of the Limit Order Book (LOB) to avoid immediate fulfillment and mislead other market participants (Lee et al., 2013; Tao et al., 2022). Traditionally, such manipulation is often detected by human experts, leading to large labor costs and low efficiency. Recently, prior research for automatic detection has explored various machine learning approaches, such as end-to-end (Cao et al., 2014; Chullamonthon & Tangamchit, 2022), or a two-stage framework (an encoder combined with a classifier) (Poutré et al., 2024; Safa et al., 2024). However, these methods often rely on level 1 tick data (i.e., the first level of LOB) anomaly modeling, overlooking manipulative behaviors that span multiple LOB levels. In practice, such multilevel manipulation strategies are not only more prevalent but also more covert and structurally complex, making them significantly harder to detect using models designed for shallow or localized patterns.

An intuitively promising direction is to utilize the structural richness of multilevel LOB data. Unfortunately, there still lacks a well-established way for the detection methods to deal with the multilevel LOB data, due to its high dimensionality, noise, and hierarchy complexity (Lu & Abergel, 2018; Marszałek & Burczyński, 2024). Hence, while manipulation detection has been approached using

diverse inputs, many methods rely on level 1 tick data or statistical indicators (Abbas et al., 2018; Rizvi et al., 2020b; Poutré et al., 2024; Safa et al., 2024; Liu et al., 2024a). Among approaches incorporating LOB, most either extract handcrafted LOB-derived features (Cao et al., 2015), use incomplete price-volume subsets (Leangarun et al., 2018; Chullamonthon & Tangamchit, 2023), or directly feed raw LOB sequences into models without explicitly modeling their hierarchical structure (Chullamonthon & Tangamchit, 2022). As a result, critical inter-level dynamics remain under-exploited. A full review of related work is provided in Appendix A.1.

To fill these gaps, we present a formalization of the multilevel manipulation detection task, according to which a two-stage framework is adopted. Then we explicitly propose a Multilevel LOB Encoder for automatically leveraging the hierarchical information in the LOB data, and subsequently concatenate the learned vectors with traditional, well-designed, manual features. This combined representation is then fed into a Contrastive Fusion Encoder, which employs supervised contrastive learning to enhance representation quality. This stage incorporates limited supervisory signals by oversampling rare anomalies and leverages a hybrid contrastive loss. In a nutshell, we design a novel framework that detects multilevel manipulation with traditional classification-based detectors, by cascading the LOB representation module and combining contrastive learning.

Building on this framework, we systematically analyze the problem of multilevel manipulation detection with extensive experiments. The results clearly reveal (i) the greater difficulty of detecting multilevel manipulation relative to single-level ones, and (ii) the tension between the informative nature of multilevel LOB structures and the inherent difficulty of leveraging them effectively. Furthermore, we show that this framework consistently improves multilevel manipulation detection performance across a variety of representation models, where Transformer-based architectures achieve the state-of-the-art results. To further demonstrate the effectiveness and generality of our framework, we conduct comprehensive ablations on the cascaded LOB representation architecture and the supervised contrastive learning component, assessing how each module and its training strategy contribute to representation quality and multilevel detection performance.

In summary, our contributions are threefold:

- We present the first method for detecting multilevel manipulation, and demonstrate its advantages and challenges over traditional single-level detection.
- We propose a novel LOB-based representation learning framework that enhances multilevel manipulation detection across diverse models, achieving state-of-the-art performance with Transformer-based architectures.
- We empirically show that LOB's hierarchical information can be effectively leveraged through representation learning, and contrastive learning brings notable gains to detection tasks.

## 2 BACKGROUND AND PROBLEM SETUP

### 2.1 LIMIT ORDER BOOK

The Limit Order Book (LOB) is a core component of the modern financial market microstructure, which serves as a dynamic electronic record of all untraded limit orders (Abergel et al., 2016). This structure is crucial for understanding market depth and liquidity, by virtue of its highly granular and deep structure and its ability to dynamically update in real time to reflect all market changes (Foucault et al., 2005).

The mathematical description of the LOB snapshot $L_t$ at any given time step $t$ can be written as:

$$L_t = \{p_a^i(t), v_a^i(t), p_b^i(t), v_b^i(t)\}_{i=1}^l.$$

Here, $l$ denotes the number of levels in the order book. For each level $i$ at time $t$, $p_a^i(t)$ and $p_b^i(t)$ represent the ask (i.e., selling) and bid (i.e., buying) prices, while $v_a^i(t)$ and $v_b^i(t)$ represent their corresponding volumes. This multilevel representation, characterized by the parameter $l$, is particularly relevant to our work, as it forms the basis for detecting multilevel manipulation.

However, the inherent complexity of LOB poses significant challenges for representation learning. First, it is high-dimensional, represented by a $4 \times l$ matrix at each time step, which requires models

capable of processing a large number of variables (Lu & Abergel, 2018). Second, it exhibits notable spatial heterogeneity, as the spread between different price levels is not constant (Gu et al., 2016). Furthermore, LOB data is characterized by both high-frequency dynamics and strong autocorrelation, as its rapid evolution reflects the complex interplay between numerous traders' actions and the market matching mechanism (Gould et al., 2013). Consequently, an effective representation is crucial, as it must account for the high-dimensional, spatial-temporal patterns in LOB data in order to detect subtle manipulative behaviors embedded across multiple levels.

## 2.2 MARKET MANIPULATION

Market manipulation is the intentional interference with market forces by an individual or group to present an unreal picture of market activity to mislead other investors for personal profit. This study focuses on Trade-Based Manipulation (TBM) (Allen & Gale, 1992) that uses real trades to execute manipulative schemes, making it difficult to detect as it seems to be legal in appearance (Khodabandehlou & Golpayegani, 2022).

Among various TBM schemes, this paper specifically delves into spoofing, which is considered one of the most covert, high-frequency, and harmful forms of abnormal trading. It is a form of market manipulation in which an individual or group places large orders with no genuine intent to execute. These orders are often submitted across multiple price levels within millisecond intervals, creating a false impression of substantial supply or demand. This misleading signal induces other investors to adjust their trading strategies accordingly. After triggering the desired market reaction, the manipulators swiftly cancel the non-*bona fide* orders and execute *bona fide* orders at more favorable prices to secure a profit (Cartea et al., 2020).

These deceptive activities leave a distinct fingerprint on the LOB, particularly at the multilevel scale (Lee et al., 2013; Tao et al., 2022). The anomalies are often hidden in deeper levels, as these orders are visible but less likely to be immediately executed, consistent with a lack of genuine trading intent. Furthermore, a key indicator is a recurring pattern where orders are placed closer to the best price to appear executable, but are canceled immediately before being filled. This cycle is repeated to influence the market without a real trading commitment.

Therefore, a model capable of effectively analyzing these multilevel, high-frequency anomalies is crucial for detecting such subtle manipulation, which motivates the design of our proposed method.

## 2.3 MULTILEVEL MANIPULATION DETECTION DEFINITION

The problem of Multilevel Manipulation Detection is formalized as a binary classification task where the goal is to determine whether a pattern of multilevel manipulation occurs at a specific time step $t$.

The input to our model is a $T$-length time-series of states, denoted as $S_t = \{X_t, \ldots, X_{t+T-1}\}$. Each $X_t$ represents the LOB snapshot $L_t \in \mathbb{R}^{4l}$ and other possible manual features $F_t \in \mathbb{R}^m$ at time $t$, where $l$ is the number of levels in LOB and $m$ is the number of manual features. The combined input is $X_t \in \mathbb{R}^{4\ell+m}$, potentially containing multilevel manipulation patterns.

The detection process is a two-stage pipeline. First, we construct a representation model $f_e$ to map the input sequence $S_t$ into a single latent feature vector $z_t$:

$$z_t = f_e(S_t), z_t \in \mathbb{R}^D, \tag{1}$$

where $D$ denotes the dimension of the latent representation. Then we build a discrimination function $g_d$ to assess whether $S_t$ contains any manipulation based on its latent representation $z_t$:

$$y_t = g_d(z_t), y_t \in \mathbb{R}, \tag{2}$$

where $y_t$ denotes the anomaly score for the sequence $S_t$, and binary predictions can be obtained by applying a threshold during evaluation. The core challenge is building a representation model that captures these intricate features to distinguish manipulation from complex market dynamics.

## 3 THE PROPOSED FRAMEWORK

Following the problem definition in Section 2.3, we adopt a two-stage framework that decouples manipulation detection into a representation stage (Eq. 1) and a subsequent anomaly detection stage

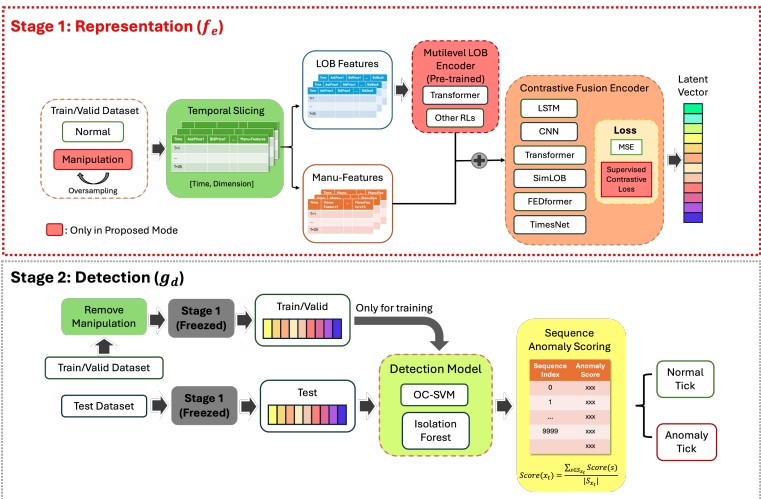

Figure 1: The overall architecture of the decoupled framework for multilevel manipulation detection, consisting of two core stages—representation and detection (Eqs. 1 and 2).

(Eq. 2). An overview of the entire framework is illustrated in Figure 1. While this architecture is common in anomaly detection, we identify representation as the key bottleneck in modeling multi-level manipulation and introduce two approaches to enhance it. The next two sections elaborate on each approach, and details of the remaining components are provided in Appendix A.2.

## 3.1 CASCADED LOB REPRESENTATION ARCHITECTURE

Effectively encoding a combination of high-dimensional LOB data and manual features requires a specialized approach, which stems from two primary factors. First, LOB data are inherently complex and highly dynamic, making it difficult for models to directly process their rich latent information without significant noise. Second, LOB data represent raw market activity, while manual features are in a processed form, creating a fundamental mismatch when these two distinct data types are simply concatenated. To address this dual challenge of LOB complexity and feature heterogeneity, we propose a cascaded representation architecture.

The first phase employs a Multilevel LOB Encoder to extract a robust latent representation from the high-dimensional LOB data. We implement this encoder using a Transformer (Vaswani et al., 2017) architecture, selected for its strong capabilities in sequence modeling. The encoder is initially pre-trained in a standalone manner to minimize the reconstruction error of the raw LOB input, formulated as the mean squared error (MSE): $\mathcal{L}_{MSE} = \frac{1}{T} \sum_{t=1}^{T} \left\| \hat{L}_t - L_t \right\|_2^2$, where $L_t$ and $\hat{L}_t$ denote the original and reconstructed LOB snapshots at time $t$, each comprising $4l$ elements corresponding to the price and volume of top $l$ bid and ask levels. The encoder is frozen after pre-training, aiming to improve training efficiency and support modular replacement with more advanced architectures.

In the second phase of our architecture, the compact latent representation produced by the Multilevel LOB Encoder is fused with the manual features. This process forms a composite feature vector that serves as the input to the Contrastive Fusion Encoder. By integrating the high-level semantic information from the LOB embedding with the structured manual features, our approach provides a comprehensive and robust representation of the market state for subsequent anomaly detection.

## 3.2 SUPERVISED CONTRASTIVE LEARNING

The inherent variability of normal market behavior, coupled with the subtle nature of multilevel manipulation, poses a fundamental challenge for traditional reconstruction-based representation models, which often fail to learn a sufficiently discriminative latent space for effective anomaly detection. To address this, we adopt a supervised contrastive learning paradigm in the representation

stage, which requires only a limited set of labeled anomalies and a modification to the loss function, resulting in a highly discriminative latent representation crucial for robust anomaly detection.

To implement this paradigm, our overall training objective of the Contrastive Fusion Encoder is a weighted combination of two complementary loss functions, defined as: $\mathcal{L} = (1 - \alpha) \cdot \mathcal{L}_{MSE} + \alpha \cdot \mathcal{L}_{SCL}$. The reconstruction loss ($\mathcal{L}_{MSE}$) serves as a foundational objective, ensuring the model learns the fundamental structure and patterns of the data, while the supervised contrastive loss ($\mathcal{L}_{SCL}$) explicitly encourages a more discriminative latent space by pulling similar samples closer and pushing dissimilar ones apart. The hyperparameter $\alpha$ is used to find the optimal trade-off between structural learning and discriminative power.

While $\mathcal{L}_{MSE}$ here extends the earlier version to reconstruct both LOB representations and manual features, our primary focus lies in $\mathcal{L}_{SCL}$ (Khosla et al., 2020), defined per training batch as:

$$\mathcal{L}_{SCL} = \frac{1}{|D|} \sum_{i \in D} - \log \frac{\sum_{j \in P(i)} e^{\text{sim}(\mathbf{z}_i, \mathbf{z}_j)/\tau}}{\sum_{k \in A(i)} e^{\text{sim}(\mathbf{z}_i, \mathbf{z}_k)/\tau}}.$$

In the formula, $\mathbf{z}$ represents the L2-normalized feature embeddings, $\text{sim}(\cdot, \cdot)$ is the cosine similarity, and $\tau$ is a temperature hyperparameter. The set $D$ contains all samples in the batch that have at least one positive pair, while $P(i)$ represents the set of positive pairs for a given anchor $i$, and $A(i)$ includes all other samples in the batch. Crucially, in the context of our anomaly detection task, a positive pair is one in which both samples are normal or anomalous, whereas a negative pair is composed of one normal sample and one anomalous.

In practice, a key challenge is the severe data imbalance, where anomalous samples are extremely rare. This is mitigated by employing an oversampling strategy during batch construction, which ensures a sufficient number of anomalous samples in each training batch for the supervised contrastive objective to operate effectively.

## 4 EXPERIMENTS

### 4.1 EXPERIMENT SETUP

#### 4.1.1 DATASET

The raw data comes from the LOBSTER platform (LOBSTER, n.d.), which has been widely used in multiple market manipulation detection investigations (Cao et al., 2015; Rizvi et al., 2020a; Safa et al., 2024; Poutré et al., 2024). It provides tick-by-tick trades and millisecond-level limit order books for multiple NASDAQ stocks. For our study, we selected three stocks representing different industries and liquidity characteristics: Cisco Systems (CSCO), Tesla (TSLA), and Intel (INTC) on January 2, 2015, with millions of entries providing sufficient data for our study. Through careful examinations, all selected data do not contain any reported market manipulation events.

Given the scarcity of real-world manipulation in high-frequency trading, we follow a widely adopted approach in both academia and industry (Cao et al., 2015) by injecting multilevel manipulation into the selected datasets. The full data processing procedure—including anomaly insertion, manual feature construction, dataset partitioning, and summary statistics—is detailed in Appendix A.3.

#### 4.1.2 BASELINES AND METRICS

To comprehensively evaluate the performance improvements enabled by the **proposed mode** (the cascaded LOB representation architecture and the combined training loss) over the **original mode** (the MSE training loss), we select a diverse set of 6 representation learning models as the Contrastive Fusion Encoder and two classic detectors for downstream evaluation. The representation models include classical architectures applied to LOB data (CNN2 (Tsantekidis et al., 2020), LSTM (Tsantekidis et al., 2017)), LOB-specific models for anomaly detection or representation learning (JFDS (Poutré et al., 2024), SimLOB (Li et al., 2024)), and state-of-the-art time-series models developed on general benchmarks (FEDformer (Zhou et al., 2022), TimesNet (Wu et al., 2023)). For downstream detectors, Isolation Forest (Liu et al., 2008) and OC-SVM (Schölkopf et al., 1999) are employed to assess the effectiveness of the learned representations and perform end-to-end comparisons on the raw data.

Table 1: Performance comparison of proposed and original modes on multilevel manipulation detection

| Detection | Representation | Mode | AUC-PR ↑ | AUROC ↑ | F4-Score ↑ | Precision ↑ | Recall ↑ |
|---|---|---|---|---|---|---|---|
| **OC-SVM** | | | 0.163 | 0.759 | 0.609 | 0.160 | 0.739 |
| | CNN2 | Original | 0.176 | 0.777 | 0.604 | 0.155 | 0.738 |
| | | Proposed | **0.198** | **0.855** | **0.707** | 0.149 | 0.923 |
| | LSTM | Original | 0.160 | 0.795 | 0.601 | 0.158 | 0.728 |
| | | Proposed | **0.375** | **0.902** | **0.734** | 0.166 | 0.935 |
| | JFDS | Original | 0.252 | 0.854 | 0.653 | 0.238 | 0.733 |
| | | Proposed | **0.675** | **0.975** | **0.881** | 0.402 | 0.952 |
| | SimLOB | Original | 0.164 | 0.768 | 0.603 | 0.144 | 0.754 |
| | | Proposed | **0.210** | **0.894** | **0.748** | 0.170 | 0.950 |
| | FEDformer | Original | **0.226** | **0.823** | 0.633 | 0.218 | 0.719 |
| | | Proposed | 0.105 | 0.787 | **0.647** | 0.106 | 0.949 |
| | TimesNet | Original | 0.186 | **0.829** | **0.611** | 0.186 | 0.713 |
| | | Proposed | **0.222** | 0.646 | 0.534 | 0.068 | 0.937 |
| **Isolation Forest** | | | 0.101 | 0.736 | 0.562 | 0.133 | 0.705 |
| | CNN2 | Original | 0.169 | 0.780 | 0.609 | 0.186 | 0.710 |
| | | Proposed | **0.209** | **0.893** | **0.732** | 0.177 | 0.911 |
| | LSTM | Original | 0.162 | 0.807 | 0.607 | 0.171 | 0.722 |
| | | Proposed | **0.364** | **0.914** | **0.750** | 0.201 | 0.904 |
| | JFDS | Original | 0.232 | 0.846 | 0.646 | 0.227 | 0.730 |
| | | Proposed | **0.631** | **0.970** | **0.855** | 0.373 | 0.930 |
| | SimLOB | Original | 0.160 | 0.779 | 0.610 | 0.163 | 0.737 |
| | | Proposed | **0.189** | **0.883** | **0.733** | 0.168 | 0.928 |
| | FEDformer | Original | 0.224 | **0.835** | 0.637 | 0.189 | 0.749 |
| | | Proposed | **0.247** | 0.817 | **0.664** | 0.114 | 0.949 |
| | TimesNet | Original | 0.187 | **0.837** | **0.625** | 0.200 | 0.721 |
| | | Proposed | **0.219** | 0.607 | 0.529 | 0.062 | 0.994 |

For performance evaluation, we employ a suite of widely-used metrics: Area Under the Precision-Recall Curve (AUC-PR), Area Under the Receiver Operating Characteristic Curve (AUROC), F-score, Recall, and Precision. Given the extreme class imbalance in our financial anomaly detection dataset, we place particular emphasis on the AUC-PR, as it provides a more reliable assessment by being sensitive to the minority class. Furthermore, as the cost of misclassifying anomalous orders is significantly higher, we utilize the F-beta measure with $\beta = 4$ to heavily weight recall and penalize false negatives.

## 4.2 EXPERIMENT 1: OVERALL PERFORMANCE EVALUATION

This set of experiments assesses our proposed representation mode to effectively exploit multilevel LOB data for multilevel manipulation detection. Both modes take 5-level LOB data and manual features as input. The original mode relies solely on MSE loss without a Multilevel LOB Encoder, while our proposed mode integrates the cascaded LOB representation architecture and supervised contrastive learning.

From Table 1, JFDS under the proposed mode achieves state-of-the-art results for both OC-SVM and Isolation Forest, demonstrating the clear superiority of our methods for multilevel manipulation detection. This significant finding is also consistent with the overall positive trend observed across most other representation models. For models such as CNN2, LSTM, and SimLOB, the proposed mode consistently leads to improvements across all evaluated metrics, which fully demonstrates its effectiveness in enhancing the representational learning capabilities. In contrast, FEDformer and TimesNet show mixed results, with some metrics improving while others decline, which implies a lack of compatibility between these general-purpose representation models and the specific characteristics of LOB data.

In conclusion, our experiments demonstrate that the proposed mode consistently enhances the performance of various representation models, except for two models for the general time series repre-

sentation learning. Furthermore, with JFDS as the Contrastive Fusion Encoder, our method successfully achieves the state-of-the-art results for the multilevel manipulation detection task.

### 4.3 EXPERIMENT 2: ANALYSIS OF THE MULTILEVEL MANIPULATION DETECTION CHALLENGE

This set of experiments investigates the challenges of multilevel manipulation detection. We first compare it with single-level manipulation to highlight its complexity, and then examine the value and challenges of using multilevel LOB data.

#### 4.3.1 COMPARISON WITH SINGLE-LEVEL MANIPULATION

As discussed in previous sections, multilevel manipulation is subtler and more prone to being overlooked or misclassified than the single-level type. To investigate this challenge more concretely, we vary the distribution of anomaly insertions and evaluate all representation models combined with OC-SVM under the original mode. Notably, we exclude multilevel LOB inputs in this setting to avoid introducing noise into reconstruction-based methods with limited capacity. We focus on the AUC-PR metric, which is particularly informative for imbalanced datasets. Results are summarized in Figure 2, with complete results in Appendix A.6.

Figure 2: Impact of anomaly insertion depth on AUC-PR.

Figure 2 reveals a consistent performance gap between models trained on single-level versus multilevel insertions, with the former achieving higher AUC-PR scores. It highlights the inherent difficulty of detecting multilevel manipulation: the anomalous signals are more dispersed across multiple levels of the order book, making them harder to localize and distinguish from normal fluctuations. These findings reinforce our hypothesis that multilevel manipulation is more complex and subtle, requiring more advanced and expressive modeling methods to detect effectively.

#### 4.3.2 THE VALUE AND CHALLENGES OF LOB REPRESENTATION

Building on earlier findings that multilevel manipulation is harder to detect, we now examine whether incorporating multilevel LOB as input improves detection performance. We evaluate three types of inputs with the OC-SVM detector for multilevel manipulation: without LOB, with raw LOB, and with embedded LOB from the Multilevel LOB Encoder. To avoid underestimating the potential of LOB modeling, we also compare the results between the two training losses of the Contrastive Fusion Encoder. We consider the AUC-PR among two groups of outputs: (i) all detected anomalies across five levels, and (ii) detected anomalies limited to levels 2–5, highlighting the model's ability to detect subtler patterns beyond level 1. Results are shown in Figure 3, with full details in Appendix A.6.

The experimental results on both metrics reveal that simply adding LOB data with MSE loss does not yield a positive performance gain. This suggests that without specialized handling, the direct inclusion of LOB data may introduce more noise, thereby underscoring the inherent challenges of LOB representation. In contrast, under our proposed combined loss, the majority of models show a significant performance improvement when using LOB or embedded LOB data, especially the latter. The only exceptions are FEDformer and TimesNet, which consistently perform better without LOB data, a finding that aligns with our conclusions from Experiment 1 regarding their incompatibility with LOB data. Furthermore, a closer look at these two metrics reveals that while proper LOB representation improves performance, the consistently lower AUC-PR evaluated in levels 2-5 compared to all-5-level reaffirms that detecting multilevel manipulation is inherently more challenging.

Overall, these results demonstrate that LOB data is indeed valuable for multilevel manipulation detection, but its effective utilization is contingent upon proper representation.

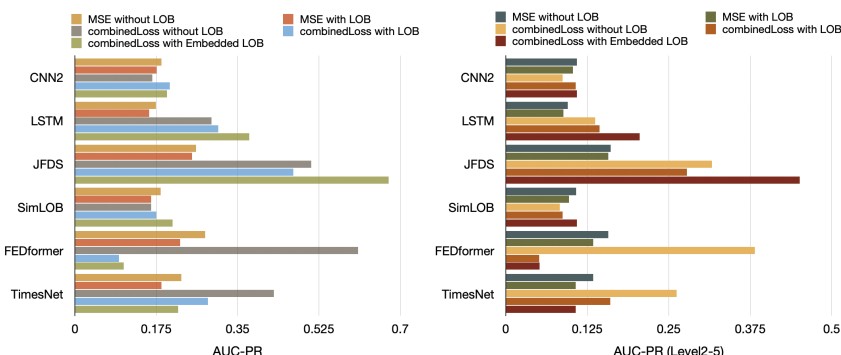

Figure 3: AUC-PR performance comparison with different loss functions and input on multilevel manipulation detection (OC-SVM): evaluated on all detected anomalies across five levels (left) and detected anomalies limited to levels 2–5 (right).

## 4.4 EXPERIMENT 3: ABLATION STUDY AND FRAMEWORK ANALYSIS

This section analyzes the representation stage of our detection framework, focusing on the cascaded architecture and the impact of supervised contrastive learning on multilevel manipulation detection.

### 4.4.1 ANALYSIS OF THE REPRESENTATION STAGE

Table 1 highlights the critical role of the representation stage in our detection framework. Across both OC-SVM and Isolation Forest, models with learned representations consistently outperform their non-representational counterparts, with the effect particularly pronounced for Isolation Forest due to its weaker native detection capability.

Examining individual architectures, Transformer-based JFDS benefits most from the proposed approaches, followed by LSTM, while CNN and SimLOB gain modestly. General-purpose models like TimesNet and FEDformer are less compatible with LOB data; in some cases, excluding incompatible inputs yields greater improvements than architectural or training changes (Figure 3).

These observations confirm that representation learning improves detection overall, but its impact differs across models, reflecting variations in architecture and compatibility with LOB data.

### 4.4.2 ABLATION STUDY OF THE CASCADED ARCHITECTURE

This experiment investigates the contribution of the Multilevel LOB Encoder in a cascaded architecture through an ablation study. We further examine how different architectural choices affect the multilevel manipulation detection performance. All settings use JFDS as the Contrastive Fusion Encoder with OC-SVM under the combined loss to ensure comparability.

As shown in Figure 4, the Multilevel LOB Encoder leads to consistent performance improvements, with the Transformer-based representation achieving the best results. While LSTM and Sim-LOB perform better, CNN2 exhibits performance degradation, suggesting that not all architectures are equally compatible for LOB representation. TimesNet and FEDformer are excluded due to their incompatibility with LOB inputs, as demonstrated in prior experiments.

Notably, the gains are most prominent in AUC-PR, with relatively smaller effects on AUROC and F4-score, indicating that the cascaded architecture is particularly effective for rare-event detection.

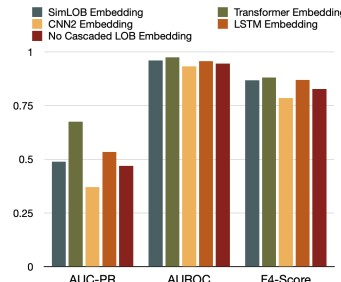

Figure 4: Ablation study on the cascaded architecture using JFDS with OC-SVM.

Table 2: Performance of JFDS with OC-SVM under varying oversampling ratios ($\beta$) using the combined loss function

| $\beta$ | AUC-PR ↑ | AUROC ↑ | F4-Score ↑ |
|---|---|---|---|
| 0.5 | 0.470 | 0.956 | 0.852 |
| 0.3 | 0.470 | 0.946 | 0.828 |
| 0.1 | 0.483 | 0.950 | 0.834 |
| 0 | - | - | - |

Table 3: Performance of JFDS with OC-SVM under different contrastive loss weight ($\alpha$)

| $\alpha$ | AUC-PR ↑ | AUROC ↑ | F4-Score ↑ |
|---|---|---|---|
| 1 | 0.252 | 0.583 | 0.498 |
| 0.8 | 0.540 | 0.962 | 0.868 |
| 0.5 | 0.526 | 0.958 | 0.851 |
| 0.2 | 0.470 | 0.946 | 0.828 |
| 0 | 0.252 | 0.854 | 0.653 |

A detailed architectural exploration of the Multilevel LOB Encoder is beyond the scope of this work and is left for future investigation.

### 4.4.3 ANALYSIS OF THE CONTRASTIVE SUPERVISED LEARNING

This section analyzes key hyperparameters in contrastive supervised learning, focusing on oversampling and loss function weighting. To isolate their effects, we evaluate JFDS with OC-SVM using raw multilevel LOB data as input, excluding the Multilevel LOB Encoder.

As shown in Table 2, the oversampling module plays a crucial role. Without it, the contrastive loss fails due to extreme class imbalance. When the anomaly ratio is set to 0.1 (i.e., anomalies comprise 10% of each batch), the model becomes consistently trainable, and further tuning has limited impact. This indicates that the presence of oversampling, rather than the precise ratio, is essential for enabling contrastive learning.

Table 3 further highlights the necessity of combining MSE and contrastive loss. Performance drops sharply when either loss is removed ($\alpha = 0$ or 1), confirming their complementarity. The contrastive loss sharpens anomaly discrimination, while MSE helps preserve structural fidelity, making the hybrid formulation critical for optimal results.

## 5 CONCLUSION

This work is the first to systematically address the challenge of detecting multilevel spoofing, a sophisticated form of trade-based manipulation, by leveraging the hierarchical information in Limit Order Book (LOB) data. We propose a representation learning framework that integrates a cascaded LOB representation architecture with supervised contrastive learning, effectively capturing complex multilevel anomaly patterns.

Experimental results demonstrate the effectiveness of our approach: our framework consistently improves detection performance across diverse models, with Transformer-based architectures achieving state-of-the-art results. We show that multilevel anomalies are inherently more subtle and challenging than single-level ones, and that LOB data, when properly represented, provides critical information for detection. Ablation studies further clarify the complementary contributions of the cascaded LOB architecture and the combined loss with limited oversampling, providing guidance for the design of robust anomaly detection.

Looking forward, future work could explore: (i) designing LOB-specific architectures for Multilevel LOB Encoder to better capture hierarchical patterns and sequential dependencies, enabling a synergistic combination of handcrafted and automatically learned features; (ii) refining the definition of supervisory signals or contrastive objectives to enhance representation quality further; and (iii) extending the framework to other types of market manipulation or more general sequential anomaly detection tasks. These directions have the potential to improve the accuracy, robustness, and applicability of the detection to various financial and sequential data scenarios.

## ETHICS STATEMENT

This work makes use of publicly available financial data from the LOBSTER platform, which provides anonymous millisecond-level limit order book and trade information for NASDAQ-listed stocks. The selected data does not include any reported cases of market manipulation and contains

no personally identifiable information. To study manipulative behaviors, we synthetically generate manipulation cases by following procedures documented in prior literature. These simulated behaviors are designed solely for defensive research purposes, and do not reflect the actions or strategies of any real market participants. Our study strictly adheres to the ICLR Code of Ethics, and we have taken necessary precautions to minimize the risk of misuse or unintended consequences.

## REPRODUCIBILITY STATEMENT

To support reproducibility, we will release the complete source code as anonymous supplementary material, including data preprocessing scripts, model implementations, and configuration files for all experimental settings. Appendix A.3 provides comprehensive details of the data preparation pipeline, including manipulation injection procedures, manual feature construction, normalization, and dataset partitioning, along with relevant parameter settings and statistics. Appendix A.4 documents the model architectures, training configurations, software environment, and implementation details for all methods used. All experiments follow a standardized training pipeline, and fair comparisons are ensured through faithful reproduction of benchmark models. We aim to make all results fully reproducible with minimal effort.

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

# A APPENDIX

## A.1 RELATED WORK

### A.1.1 ANOMALY DETECTION OF MARKET MANIPULATION

Early studies on market manipulation detection are primarily rule-based or statistical (Jarrow, 1992; Kirkland et al., 1999; Aggarwal & Wu, 2006; Mongkolnavin & Tirapat, 2009), relying heavily on expert-defined heuristics or handcrafted indicators. While interpretable, these methods suffer from poor adaptability and generalization, making them ineffective against evolving or subtle manipulation strategies (Chullamonthon & Tangamchit, 2023).

To overcome these limitations, classical machine learning techniques such as support vector machines and decision trees have been explored (Öğüt et al., 2009; Diaz et al., 2011). A representative example is the Adaptive Hidden Markov Model with Anomaly States (AHMMAS) proposed by (Cao et al., 2015), which achieves improved performance over previous methods by modeling transitions between normal and anomalous states. However, AHMMAS suffers from exponential growth in computational complexity as the number of input features increases, making it difficult to scale. In general, these methods remain limited in capturing complex patterns in high-dimensional settings, motivating the adoption of deep learning approaches.

To better analyze recent deep learning efforts in manipulation detection, we organize existing methods into two main categories: end-to-end models and autoencoder-based two-stage frameworks. While the former directly learns decision boundaries from raw inputs, the latter focuses on extracting informative embeddings to support downstream detection.

Regarding end-to-end models, one study employs a multilayer perceptron (MLP) to detect synthetic pump-and-dump patterns from level-1 data (Leangarun et al., 2016), while another uses a

Transformer-based classifier to improve detection on both synthetic and real-world cases by capturing richer temporal dependencies (Chullamonthon & Tangamchit, 2022). These methods demonstrate the ability of deep learning to model complex dependencies in high-dimensional data without relying on handcrafted features. However, most end-to-end approaches rely on fully supervised training, which requires large volumes of labeled data that are often unavailable in real markets. In addition, they generally lack adaptability, as a separate model must be retrained from scratch to handle each new type of manipulation.

In parallel, two-stage methods aim to extract informative representations from market data, typically trained in an unsupervised manner and used in conjunction with downstream classifiers for anomaly detection. An LSTM-based autoencoder detects manipulation in the Thai market using reconstruction error and shows superior performance over an LSTM-GAN in capturing pump-and-dump (Leangarun et al., 2021). Another approach learns representation using affinity matrices, with manipulation detected via kernel density–based clustering, showing notable improvements on LOB-STER data (Rizvi et al., 2020a). WALDATA transforms stock price time series into 2D scalogram images using wavelet transforms and applies a GAN to learn normal trading behavior, with the discriminator detecting manipulation (Safa et al., 2024). A transformer encoder is also explored to extract representations from high-frequency LOB data, with an OC-SVM applied to identify manipulation (Poutré et al., 2024). Overall, two-stage frameworks can extract informative representations from high-dimensional market data, enabling downstream detection methods that would otherwise struggle with such inputs. In addition, they reduce reliance on labeled data compared to fully supervised models and allow greater flexibility for adapting to new manipulation types with lower retraining cost.

Although these deep learning approaches achieve notable results, they primarily focus on single-level anomalies and often overlook covert manipulative behaviors that span multiple LOB levels. Such cross-level manipulations are both structurally complex and widely distributed, making them difficult to detect with conventional methods. However, high-frequency multilevel LOB data encodes rich hierarchical signals that can be critical for identifying these subtle patterns. To this end, two-stage representation learning offers a natural solution, as it is well-suited for capturing structure in high-dimensional, noisy, and unlabeled data. Motivated by these strengths, we explore a two-stage framework tailored to LOB, aiming to improve the detection of multilevel market manipulation.

### A.1.2 REPRESENTATION LEARNING FOR LOB

Representation learning plays a central role in modeling multivariate time series (MTS), enabling the extraction of compact and informative features from noisy, high-dimensional, and non-stationary sequences (Zhang et al., 2024). This capability supports a wide range of downstream tasks such as classification (Middlehurst et al., 2024), forecasting (Cai et al., 2024), and anomaly detection (Choi et al., 2021), and has become fundamental in many domains, including finance, healthcare, and industrial systems (Trirat et al., 2024). Among them, Limit Order Book (LOB) data represents a particularly complex form of MTS—characterized by high dimensionality, spatial heterogeneity, and rapid temporal dynamics—making effective representation learning especially critical for downstream modeling.

Recent progress in time series representation learning has led to a diverse set of architectures designed to model multivariate temporal dependencies. MLP-based models such as TimeMixer (Wang et al., 2024a) exploit structured mixing over time and features; convolutional approaches like Times-Net (Wu et al., 2023) leverage hierarchical receptive fields to capture multi-scale patterns; recurrent frameworks such as Mamba (Gu & Dao, 2024) introduce state-space modeling for long-range dynamics; and Transformer variants, including FEDformer (Zhou et al., 2022), PatchTST (Nie et al., 2023), and iTransformer (Liu et al., 2024b), enable efficient sequence modeling with enhanced scalability and global context integration. While these models have achieved state-of-the-art results across forecasting and classification benchmarks, they are often developed with general-purpose or task-specific objectives, and their direct applicability to domain-specific settings such as LOB modeling remains limited due to the latter's unique structural properties (Zhong et al., 2025).

Meanwhile, several studies have developed models specifically tailored for LOB data. CNN2 (Tsantekidis et al., 2020) and LSTM (Tsantekidis et al., 2017) serve as foundational baselines, with CNN2 leveraging convolutional filters to extract local features and LSTM capturing sequential dependencies. DeepLOB (Zhang et al., 2019) integrates a CNN module for spatial feature extraction with

an LSTM layer to model temporal dynamics, effectively handling the high-frequency volatility and sequential structure of LOB data. TransLOB (Wallbridge, 2020) further incorporates a Transformer encoder to capture long-range dependencies, with CNNs modeling short-term fluctuations. More recently, SimLOB (Li et al., 2024) adopts a Transformer-based encoder-decoder architecture, applying fully connected layers before and after attention modules to enhance representation capacity. By reconstructing LOB sequences from latent embeddings, it emphasizes representation learning more explicitly.

However, most existing LOB-specific models remain task-specific and end-to-end, typically designed for applications such as price forecasting or market simulation. Even the approach with explicit representation learning objectives, like SimLOB, is generally oriented toward calibration rather than manipulation detection. As a result, there remains a significant gap in leveraging LOB representations for market manipulation detection, where the structural complexity of multilevel LOBs demands more flexible, representation-centric modeling approaches.

## A.2 FRAMEWORK DETAILS

In Section 3 of the main text, we introduce the core innovations of our framework, including the cascaded LOB representation architecture and supervised contrastive learning. The overall structure is illustrated in Figure 1, which serves as a reference throughout this section. In this appendix, we provide additional details from the perspective of the general architecture, offering a more comprehensive explanation of each component and its integration into the overall framework.

### A.2.1 REPRESENTATION

The representation stage in our framework is designed to compress complex, high-dimensional market data into compact latent vectors suitable for downstream anomaly detection.

This stage takes both raw LOB data and a set of manual features as input. Unlike conventional methods that typically construct training sets using only normal data, we incorporate a small portion of labeled anomalies and perform oversampling to address extreme class imbalance. The resulting dataset is then partitioned into overlapping time-series sequences via a sliding-window mechanism. These sequences are subsequently processed by a configurable autoencoder-based module, which serves as the core of our representation stage.

Within the core module, our framework extends the traditional autoencoder-based representation learning paradigm in two key ways. First, instead of directly concatenating raw LOB data with manual features as in conventional methods, we introduce a Multilevel LOB Encoder that separately encodes LOB inputs to extract hierarchical information before combining them with manual features. Second, rather than relying solely on a reconstruction loss (e.g., MSE), we employ a hybrid training objective in the Contrastive Fusion Encoder that integrates supervised contrastive learning with reconstruction, thereby improving the discriminative quality of the learned latent space. The final latent vector is obtained from the Contrastive Fusion Encoder's output and used for downstream anomaly detection.

It is worth noting that the process described above corresponds to the training phase. During inference, labels are no longer required: the pretrained and frozen representation stage directly transforms incoming sequences into latent vectors for use by the downstream detection module.

### A.2.2 ANOMALY DETECTION

The final stage of our framework is the anomaly detection module, which operates on the latent vectors produced by the frozen representation stage. Its goal is to identify whether each market behavior is normal or manipulated.

During training, we adopt unsupervised learning by fitting a detector—such as OC-SVM (Schölkopf et al., 1999) or Isolation Forest (Liu et al., 2008)—on latent vectors derived exclusively from normal data of the training set used in the previous stage. This approach allows the model to learn the underlying distribution of typical market dynamics without relying on scarce anomaly labels. We choose these detectors for their compatibility with high-dimensional latent spaces and their computational efficiency, which makes them preferable to fully end-to-end alternatives in this context.

At inference time, new data are first passed through the same frozen representation stage to obtain latent vectors. These are then evaluated by the trained detection model to produce segment-level anomaly scores. To generate point-wise anomaly scores for each time step, we aggregate overlapping window predictions following the method in (Poutré et al., 2024). Any time step with a score exceeding a predefined threshold is then labeled as manipulated.

## A.3 DATA PREPARATION

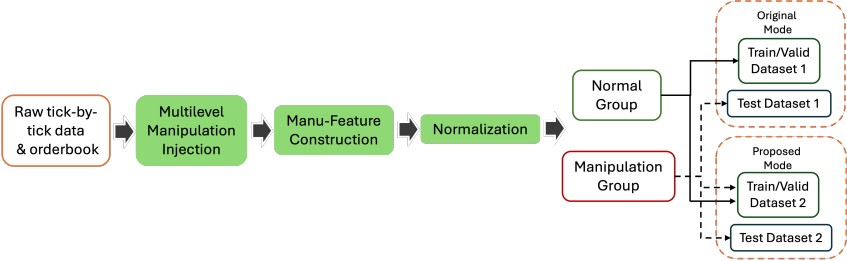

Figure 5: The overall pipeline of data preparation for the multilevel manipulation detection task.

To support the novel task of multilevel manipulation detection, we construct a comprehensive and challenging dataset from raw limit order book (LOB) snapshots and tick-by-tick transaction records. The overall data preparation pipeline consists of four key phases: multilevel manipulation injection, manual feature construction, normalization, and dataset partitioning. An overview of the pipeline is shown in Figure 5.

Our approach to manipulation injection is particularly noteworthy as it diverges from prior studies. Rather than targeting a single LOB level, we inject manipulation events across all five levels, following empirically derived distribution patterns to better reflect realistic behavior. A detailed description of the injection process and parameter configurations is provided in Appendix A.3.1.

Following the manipulation injection, we construct a set of manual features derived from both LOB and transaction-level information. These features, commonly used in related work, are intended to complement the raw LOB input and enhance the interpretability and performance of the model. A complete description of these features is provided in Appendix A.3.2. All input features are then standardized using Z-score normalization to facilitate stable model training.

Finally, we adopt two dataset partitioning approaches aligned with different experimental setups. The standard partitioning approach, used for conventional reconstruction-based methods, trains only on normal data. In contrast, our proposed setup includes a small fraction of labeled anomalies in the training set—critical for enabling supervised contrastive learning discussed in Section 3. Full statistics for each dataset are also summarized in Appendix A.3.2.

### A.3.1 MANIPULATION INSERTIONS

Given a sequence of LOB snapshots $\{L_t, \ldots, L_{t+k}\}$, we simulate multilevel synthetic manipulation by injecting anomalous patterns into the sequence, which is adapted from the work of (Poutré et al., 2024). These patterns are designed to emulate spoofing/layering strategies commonly observed in real-world financial markets. We detail the insertion procedure using bid-side as an example; the ask-side counterpart follows the same logic with reversed directionality. The full insertion procedure is outlined below, and the specific parameter configurations used in our simulation are summarized in Table 4.

1. **Index Selection:**
   Candidate time steps $t$ are selected as the start of manipulation if they satisfy the following condition:
   $$\frac{\text{BidPrice1}(t)}{\text{AskPrice1}(t)} \geq 1.0008$$
   Additionally, $t$ must not be within a window of existing anomalies to avoid interference between events.

Table 4: Parameters for multilevel manipulation injection

| Parameter | Bona Fide | Non-Bona Fide |
|---|---|---|
| Order Side | {Bid, Ask} | {Bid, Ask} |
| Order Price | Price1 $\pm$ (0, 3) bps | (Prize5, Price1 $\pm$ (0, 7) bps] |
| Total Volume | $[2, 3] \times$ Avg. order size | $[5, 6] \times$ Avg. order size |
| Number of Orders | 1 | [10, 15] |
| Interarrival time | [1, 5] ms | [10, 20] ms |
| Cancellation Delay | - | [100, 500] ms |
| Trade Delay | [10, 20] ms | - |

2. **Insertion of a *Bona Fide* Order:**
   At 1–5 ms after time $t$, a *bona fide* order is inserted on the ask side, typically priced 0–3 bps below the best ask price and sized at 2–3 times the average order volume. This order reflects the manipulator's true trading intent and is expected to be executed during the manipulation.

3. **Placement of Non-*Bona Fide* Orders:**
   A sequence of 10–15 non-*bona fide* orders is placed on the bid side, spanning LOB levels 5 to 1. These orders are submitted at progressively higher prices, uniformly distributed between the BidPrice5 and BidPrice1 plus 0.7 bps, with each order spaced 10–15 ms apart. All orders have equal volume, and the total volume of the sequence is scaled to approximately 5–6 times the average order volume. The intent is to create a deceptive appearance of strong buying pressure, thereby influencing other participants to adjust their orders or market expectations in response to the perceived demand.

4. **Execution of the *Bona Fide* Order:**
   Approximately 10–20 ms after the non-*bona fide* sequence is initiated, market participants begin reacting to the apparent demand. As a result, the previously placed *bona fide* order is fully executed, allowing the manipulator to complete a favorable transaction.

5. **Cancellation of Non-*Bona Fide* Orders:**
   After the *bona fide* order is executed, the manipulator waits approximately 100–500 ms before canceling all non-*bona fide* orders in a single batch. This delayed cancellation helps avoid unintentional fulfillment and marks the completion of the manipulation operation.

### A.3.2 DATASET STATISTICS AND INPUT FEATURES

To complement the raw LOB input, we incorporate a set of manual features into the representation stage. These features are selected based on their widespread use in prior studies on single-level anomaly modeling, allowing for a fair and consistent comparison with existing approaches. They also provide a structured way to incorporate domain knowledge, helping the model to better capture indicative market behaviors.

The selected features can be grouped into four categories, including return-based dynamics, trade and cancellation volumes, event indicators, and time intervals between market events. A complete list of these manual features is summarized in Table 5.

With the feature representation defined, we next detail the overall dataset statistics used in our experiments. To support the evaluation of our framework, we prepare two versions of the dataset under different training configurations: the original setting, which includes only normal data in the training set (commonly used in reconstruction-based methods), and our proposed setting, which includes a small proportion of labeled anomalies to enable supervised contrastive learning. Both versions share the same testing dataset but differ in training/validation.

Table 6 summarizes the data distribution across the training, validation, and testing splits under both settings. Notably, the proposed training mode maintains a highly imbalanced structure, with anomalies comprising only 0.03% of the training set. At the same time, the total number of orders exceeds 1.2 million in the training set alone, providing sufficient scale to support representation learning on high-dimensional LOB data.

Table 5: List of manual features for manipulation detection

| Feature | Description |
|---|---|
| ReturnBid1 | Best bid-price return at event $t$ |
| ReturnAsk1 | Best ask-price return at event $t$ |
| DerivativeReturnBid1 | Difference quotient of best bid-price return *w.r.t.* time, at event $t$ |
| DerivativeReturnAsk1 | Difference quotient of best ask-price return *w.r.t.* time, at event $t$ |
| TradeBidSize | Moving average of trade size consuming liquidity at best bid |
| TradeAskSize | Moving average of trade size consuming liquidity at best ask |
| CancelledBidSize | Moving average of cancellation size at best bid-price |
| CancelledAskSize | Moving average of cancellation size at best ask-price |
| TradeBidIndicator | Indicator of trade rapidity at best bid-price |
| TradeAskIndicator | Indicator of trade rapidity at best ask-price |
| CancelledBidIndicator | Indicator of cancellation rapidity at best bid-price |
| CancelledAskIndicator | Indicator of cancellation rapidity at best ask-price |
| DeltaTime | the time delta between market events $t$ and $t-1$ |

Table 6: Distribution of training, validation, and testing sets under original and proposed training modes

| Dataset | Training Mode | Total Orders | Manipulated Orders | Anomaly Ratio (%) |
|---|---|---|---|---|
| Training | Original | 1254707 | 0 | 0.00 |
| | Proposed | 1239632 | 388 | 0.03 |
| Validation | Original | 295482 | 0 | 0.00 |
| | Proposed | 324807 | 74 | 0.02 |
| Testing | Original/Proposed | 66524 | 3350 | 5.04 |

## A.4   IMPLEMENTATION DETAILS

All experiments are implemented in Python 3.12 using PyTorch 2.6.0 (Paszke et al., 2019) and Lightning 2.5.0 [1]. Training is conducted on a workstation equipped with dual NVIDIA RTX A5000 GPUs (24GB each), with experiment management and logging handled via the Comet [2] platform.

To ensure consistent evaluation across models, we adopt a unified training pipeline with a fixed sequence length of 25 and a batch size of 256. Each model is optimized using Adam (Kingma & Ba, 2015) with a learning rate of $1 \times 10^{-4}$ for 10 epochs.

All baseline models are faithfully adapted from the standardized Time Series Library (Wang et al., 2024b), which provides standardized implementations of a wide range of deep time series models. For models not included in this benchmark, we follow the original official code. All model structures and hyperparameters are kept consistent with the original implementations unless otherwise specified, ensuring a fair and reproducible comparison.

## A.5   THE USE OF LARGE LANGUAGE MODELS (LLMs)

We used large language models (LLMs), specifically ChatGPT, as a writing assistant to improve the fluency and clarity of English expressions throughout the paper. This includes grammar corrections, sentence rephrasings, and consistency adjustments. The LLM was not involved in any part of the research process, such as ideation, experimental design, data analysis, literature review, or content generation. All substantive content and scientific contributions were conceived and developed by the authors. The authors bear full responsibility for the accuracy and integrity of the content.

---

[1]`https://lightning.ai`
[2]`https://www.comet.com`

Table 7: Performance comparison with different anomaly insertion depths (OC-SVM) without LOB input under the original mode

| Representation | Anomaly Insertion | AUC-PR ↑ | AUROC ↑ | F4-Score ↑ |
|---|---|---|---|---|
| CNN2 | 1 level | **0.290** | 0.747 | 0.533 |
|  | 5 levels | 0.186 | **0.777** | **0.606** |
| LSTM | 1 level | **0.262** | **0.839** | 0.613 |
|  | 5 levels | 0.174 | 0.837 | **0.622** |
| JFDS | 1 level | **0.308** | 0.868 | 0.648 |
|  | 5 levels | 0.260 | **0.889** | **0.680** |
| SimLOB | 1 level | **0.274** | 0.736 | 0.541 |
|  | 5 levels | 0.184 | **0.793** | **0.605** |
| FEDformer | 1 level | **0.396** | 0.891 | 0.697 |
|  | 5 levels | 0.280 | **0.913** | **0.762** |
| TimesNet | 1 level | 0.222 | 0.853 | **0.638** |
|  | 5 levels | **0.229** | **0.857** | 0.633 |

## A.6 FULL RESULTS

This section provides the full results for Experiment II, which are partially reported in Section 4. Specifically, Table 7 reports the performance under different anomaly insertion depths, while Table 8 and Table 9 summarize the effects of different loss functions and input modes for multilevel manipulation detection. These results complement our analysis in the main text by offering a more detailed view of the evaluation.

Table 8: Performance comparison with different loss functions and input on multilevel manipulation detection (OC-SVM): all-5-levels anomalies

| Representation | Loss | LOB | AUC-PR ↑ | AUROC ↑ | F4-Score ↑ |
|---|---|---|---|---|---|
| CNN2 | MSE | No | 0.186 | 0.777 | 0.606 |
| | MSE | Yes | 0.176 | 0.777 | 0.604 |
| | combinedLoss | No | 0.166 | 0.735 | 0.540 |
| | combinedLoss | Yes | **0.204** | **0.874** | **0.770** |
| | combinedLoss | Yes (Embed) | 0.198 | 0.855 | 0.707 |
| LSTM | MSE | No | 0.174 | 0.837 | 0.622 |
| | MSE | Yes | 0.160 | 0.795 | 0.601 |
| | combinedLoss | No | 0.294 | **0.927** | **0.805** |
| | combinedLoss | Yes | 0.308 | 0.910 | 0.782 |
| | combinedLoss | Yes (Embed) | **0.375** | 0.902 | 0.734 |
| JFDS | MSE | No | 0.260 | 0.889 | 0.680 |
| | MSE | Yes | 0.252 | 0.854 | 0.653 |
| | combinedLoss | No | 0.508 | 0.952 | 0.861 |
| | combinedLoss | Yes | 0.470 | 0.946 | 0.828 |
| | combinedLoss | Yes (Embed) | **_0.675_** | **_0.975_** | **_0.881_** |
| SimLOB | MSE | No | 0.184 | 0.793 | 0.605 |
| | MSE | Yes | 0.164 | 0.768 | 0.603 |
| | combinedLoss | No | 0.164 | 0.809 | 0.669 |
| | combinedLoss | Yes | 0.175 | 0.841 | 0.722 |
| | combinedLoss | Yes (Embed) | **0.210** | **0.894** | **0.748** |
| FEDformer | MSE | No | 0.280 | 0.913 | 0.762 |
| | MSE | Yes | 0.226 | 0.823 | 0.633 |
| | combinedLoss | No | **0.609** | **0.966** | **0.862** |
| | combinedLoss | Yes | 0.095 | 0.769 | 0.660 |
| | combinedLoss | Yes (Embed) | 0.105 | 0.787 | 0.647 |
| TimesNet | MSE | No | 0.229 | 0.857 | 0.633 |
| | MSE | Yes | 0.186 | 0.829 | 0.611 |
| | combinedLoss | No | **0.428** | **0.909** | **0.752** |
| | combinedLoss | Yes | 0.286 | 0.690 | 0.507 |
| | combinedLoss | Yes (Embed) | 0.222 | 0.646 | 0.534 |

Table 9: Performance comparison with different loss functions and input modes on multilevel manipulation detection (OC-SVM): anomalies detected in levels 2-5 only

| Representation | Loss | LOB | AUC-PR L2-5 ↑ | AUROC L2-5 ↑ | F4-Score L2-5 ↑ |
|---|---|---|---|---|---|
| CNN2 | MSE | No | **0.109** | 0.797 | 0.542 |
| | MSE | Yes | 0.103 | 0.800 | 0.530 |
| | combinedLoss | No | 0.087 | 0.711 | 0.440 |
| | combinedLoss | Yes | 0.107 | **0.863** | **0.629** |
| | combinedLoss | Yes (Embed) | **0.109** | 0.851 | 0.558 |
| LSTM | MSE | No | 0.095 | 0.848 | 0.541 |
| | MSE | Yes | 0.088 | 0.818 | 0.526 |
| | combinedLoss | No | 0.137 | **0.916** | **0.661** |
| | combinedLoss | Yes | 0.144 | 0.899 | 0.640 |
| | combinedLoss | Yes (Embed) | **0.205** | 0.897 | 0.611 |
| JFDS | MSE | No | 0.161 | 0.899 | 0.621 |
| | MSE | Yes | 0.157 | 0.876 | 0.623 |
| | combinedLoss | No | 0.316 | 0.948 | 0.754 |
| | combinedLoss | Yes | 0.278 | 0.943 | 0.721 |
| | combinedLoss | Yes (Embed) | **0.451** | **0.973** | **0.811** |
| SimLOB | MSE | No | 0.108 | 0.821 | 0.533 |
| | MSE | Yes | 0.097 | 0.800 | 0.534 |
| | combinedLoss | No | 0.083 | 0.789 | 0.528 |
| | combinedLoss | Yes | 0.087 | 0.824 | 0.572 |
| | combinedLoss | Yes (Embed) | **0.109** | **0.889** | **0.600** |
| FEDformer | MSE | No | 0.157 | 0.910 | 0.671 |
| | MSE | Yes | 0.134 | 0.847 | 0.592 |
| | combinedLoss | No | **0.382** | **0.960** | **0.762** |
| | combinedLoss | Yes | 0.051 | 0.780 | 0.484 |
| | combinedLoss | Yes (Embed) | 0.052 | 0.784 | 0.483 |
| TimesNet | MSE | No | 0.134 | 0.870 | 0.576 |
| | MSE | Yes | 0.107 | 0.845 | 0.563 |
| | combinedLoss | No | **0.262** | **0.902** | **0.607** |
| | combinedLoss | Yes | 0.16 | 0.665 | 0.342 |
| | combinedLoss | Yes (Embed) | 0.107 | 0.615 | 0.360 |

