# OpenReview forum: "Detecting Multilevel Manipulation from Limit Order Book via Cascaded Contrastive Representation Learning"
_ICLR.cc/2026/Conference — ICLR 2026 Conference Withdrawn Submission_

### Official Review · Reviewer_5FP8 · 2025-10-27

**Soundness:** 2
**Presentation:** 2
**Contribution:** 2
**Rating:** 4
**Confidence:** 2

**Summary:**

The paper addresses trade-based manipulation detection, specifically spoofing, within limit order book (LOB) data. It formalises “multilevel manipulation” as anomalies across multiple LOB levels and proposes a two-stage representation learning framework: a Cascaded LOB Encoder to capture hierarchical structure, followed by a Contrastive Fusion Encoder trained with a hybrid loss combining MSE and supervised contrastive learning. The approach is evaluated on synthetic LOBSTER data with injected manipulations, showing performance improvements across several LOB representation models.

**Strengths:**

- The paper structure is clear and easy to follow.

- The paper identifies multilevel rather than single-level manipulation.

**Weaknesses:**

- The key novelty is a bit unclear. The overall architecture is well-known. The “cascaded” design is a concatenation of LOB embeddings and manual features. The contrastive component follows Khosla et al. (2020) directly, without any modification for the financial or LOB context, which does not fully support the claim of "novel LOB-based representation learning framework".

- The paper would be benefited from further analysis about the interpretation on what features or inter-level interactions matter most.

- It seems there are some missing baselines. For example DeepSAD, Deep Isolation Forest, FeaWAD etc.

Ruff, Lukas, et al. "Deep semi-supervised anomaly detection." arXiv preprint arXiv:1906.02694 (2019).

Xu, Hongzuo, et al. "Deep isolation forest for anomaly detection." IEEE Transactions on Knowledge and Data Engineering 35.12 (2023): 12591-12604.

Tan, Xu, et al. "Weakly-supervised anomaly detection for multimodal data distributions." 2024 IEEE International Conference on Signal Processing, Communications and Computing (ICSPCC). IEEE, 2024.

- The latency analysis is missing. As this paper focuses on LOB data, the application would be on the high-frequency trading scenario where decisions must be near-real-time, but no runtime or throughput analysis is reported.

**Questions:**

- Is it possible to involve the missing baselines mentioned above?

- What's the latency of the proposed framework?

---

### Official Review · Reviewer_L9LQ · 2025-10-30

**Soundness:** 2
**Presentation:** 2
**Contribution:** 2
**Rating:** 2
**Confidence:** 4

**Summary:**

This paper introduces a framework for detecting multilevel spoofing by leveraging hierarchical information from the Limit Order Book (LOB). The authors propose a two-stage cascaded representation learning framework, combining (1) a Multilevel LOB Encoder (Transformer-based, pre-trained with MSE reconstruction) and (2) a Contrastive Fusion Encoder trained with a hybrid loss (MSE + supervised contrastive). The approach aims to enhance discriminative representation learning for anomaly detection. Experiments on synthetic manipulation data injected into LOBSTER datasets (CSCO, TSLA, INTC) demonstrate consistent improvements across several baseline architectures (CNN2, LSTM, JFDS, SimLOB, FEDformer, TimesNet). The authors claim state-of-the-art results for multilevel manipulation detection and provide ablations and analysis of key components.

**Strengths:**

- The motivation is well written, and it is easy to know what problem this work tends to solve (detecting multilevel spoofing), making the research direction well-motivated
- The paper conducts a comprehensive experimental evaluation across 6 different representation models and 2 downstream detectors, systematically comparing the proposed mode against a baseline
- The paper provides thorough ablation studies and sensitivity analyses to validate its components , including the contribution of the cascaded LOB encoder and the impact of contrastive learning parameters like oversampling and loss weighting

**Weaknesses:**

- The entire evaluation relies on *synthetic manipulations* injected into LOBSTER data. While the procedure is documented, it remains unknown whether these patterns realistically mimic genuine spoofing. No validation or sensitivity analysis is provided comparing synthetic and real manipulative behaviors. This makes empirical conclusions about “state-of-the-art performance” somewhat weak.
- The proposed framework largely builds upon existing paradigms, specifically, a stacked autoencoder combined with supervised contrastive learning. While the paper presents these components in a cascaded form and applies them to a novel domain, the underlying algorithmic ideas are not fundamentally new. The contribution lies primarily in system integration and empirical validation rather than in advancing the methodological frontier of representation learning or anomaly detection.
- The paper's methodological pipeline, particularly the detection stage, lacks sufficient detail. The authors should provide a clearer, more formal description (including mathematical formulations) of the complete workflow, from data preprocessing through both the representation and detection stages.
- The comparison between the "proposed mode" and "original mode" is confounded, making it difficult to attribute performance gains. The "proposed mode" bundles multiple changes (a new cascaded architecture, a hybrid contrastive loss, and an oversampling strategy) , and the paper fails to disentangle the individual impact of each component relative to the simpler "original mode" baseline.
- The paper lacks any analysis or visualization of the learned representations, which makes it difficult to assess what the model has captured.
- Experimental reliability is questionable since all reported results appear to be obtained from a single random seed, raising concerns about statistical robustness.

**Questions:**

- According to your methodological description, the Transformer model is first pre-trained as a LOB Encoder. Wouldn’t this make the approach a three-phase pipeline (pre-training, contrastive fusion training, and detection) rather than a two-phase one?
- Since contrastive learning relies on in-batch samples, how does the batch size influence model performance?
- As the manipulated data are synthetically generated by the authors, could you clarify the rationale for applying oversampling rather than generating additional manipulated samples?
- Could you include comparisons with more recent and stronger time series models (e.g., TimeMixer, Mamba, etc.) as benchmarks, given that FEDformer and TimesNet are both from 2022?
- Why is the ablation of “MSE with Embedded LOB” missing from Figure 3 and Tables 8/9? This appears to be an important control experiment to isolate the effect of the combined loss.
- The entire evaluation depends on *synthetically injected manipulations*. How can we be confident these simulations faithfully represent real spoofing dynamics?

---

### Official Review · Reviewer_37Bp · 2025-11-01

**Soundness:** 2
**Presentation:** 2
**Contribution:** 2
**Rating:** 4
**Confidence:** 2

**Summary:**

This paper proposes a representation learning framework combining a cascaded LOB representation architecture with supervised contrastive learning to detect multilevel trade-based manipulation from Limit Order Book data. The framework leverages hierarchical information across multiple price levels using Transformer-based architectures, achieving consistent improvements in detection performance across diverse models. Extensive experiments and ablation studies further demonstrate its effectiveness and provide insights into representation learning and anomaly detection for complex time series data.

**Strengths:**

1. The paper introduces a well-motivated and technically sound framework that effectively captures hierarchical information in the Limit Order Book through a cascaded Transformer-based representation architecture.
2. It demonstrates consistent and meaningful performance improvements across multiple baseline models, supported by extensive experiments and systematic ablation studies.
3. The work provides valuable insights into how contrastive representation learning can enhance anomaly detection in complex multilevel financial time series data.

**Weaknesses:**

1. The performance improvement is inconsistent, and in several cases the proposed method even leads to performance drops compared with baselines.
2. The novelty is limited; as a task-specific approach that extends standard time-series modeling with an additional hierarchical dimension, the proposed design feels like a natural rather than a fundamentally new solution.
3. The evaluation relies on only one dataset. Although the dataset is sufficiently large, testing on additional datasets would be necessary to demonstrate robustness and generalizability.

**Questions:**

See Weaknesses.

---

### Official Review · Reviewer_ADis · 2025-11-02

**Soundness:** 2
**Presentation:** 4
**Contribution:** 2
**Rating:** 2
**Confidence:** 3

**Summary:**

This paper addresses the problem of spoofing detection on multi-level limit order book (LOB) data.
The authors propose a number of improvements over previous approaches, namely:
- Relying on multi-level order book data, rather than single level
- Introducing a transformer-based encoder to extract meaningful representations from the temporal sequence of LOB data
- Introducing a supervised contrastive term in the loss of an overall feature encoder to be used for downstream anomaly detection

**Strengths:**

- The paper is well written. Everything is very clearly explained, with additional details available in comprehensive appendices. Motivation and background are well sourced with many citations supporting the main claims.
- The proposed improvements are interesting and seems to meaningfully improve upon previous work in this domain.
- Results show that the proposed encoding for multi-level LOB data significantly improves detection when used in conjunction with previous state of the art contrastive fusion encoder approaches.
- The paper includes ablations for all of the different proposed improvements over previous SOTA approaches as well as analyses of impact of key hyperparameters.

**Weaknesses:**

The following are, in my opinion, the main weaknesses of the paper, roughly in order of importance.

### 1. Scope and Novelty

This is fundamentally an applied paper that would seem to be more suitable for a conference with an applied track or a domain specific conference or journal. While the paper is well written, the problem being solved has a very narrow scope within a specific application domain and the proposed solution, while interesting, does not innovate by proposing more generally applicable architectural components. It merely composes well established building blocks into a coherent architecture to solve the problem.

### 2. Synthetic Anomalies

The experiments rely on injected synthetic spoofing events rather than real observed events. While the authors argue that this is commonplace in this domain, it still significantly subtracts from the presented results in my view. Particularly given that previous approaches don't seem to rely on supervised labels for training the encoder while the present work does. The fact that the authors use the same anomaly generating process for training and test therefore raises questions. Including a robustness analysis to "model mis-specification", with test anomalies following a process with different assumptions than the model was trained on could improve this. However, sourcing real anomalies would still be the definitive test.

### 3. Missing sensible supervised baselines

Given that the authors are already assuming access to labelled data for their proposed supervised contrastive loss, why not try a supervised approach directly? While I have no experience in this particular application domain, in fraud detection, which suffers from the same problem of very low incidence of fraud examples, supervised learning approaches are typically preferred, at least when labels are available. As such, it is strange to me that a supervised baseline trained end-to-end for binary cross-entropy is not provided as a point of comparison as it would be a much simpler solution, without requiring separate representation learning and anomaly detection steps.

### 4. Lack of error bars

All experiments seem to have been carried out using a single run. Multiple runs with different seeds for the anomaly injection process and the model initialization (plus any other relevant stochastic input to the process) should be used instead. Particularly since the proposed method is relies on contrastive learned embeddings followed by anomaly detection which, in my experience, can be very noisy. In particular, the results for many of the ablations seem very inconclusive and error bars are a necessity to understand if the effects are potentially just noise.

### Minor Points

The order book definition is not very precise. I assume that the levels are defined such that, for example, $p_b^1 > p_b^2 > ... > p_b^{i+1}$. That is, that the first level corresponds to the lowest ask and the highest bid. A diagram depicting a typical order book with the labeled quantities (i.e., the bids, asks and the volumes) would be helpful.

I also wonder why the authors are defining the order book like this, with a joint level for bids and asks? Are these sides not independent, potentially having different depths?

**Questions:**

Mostly my questions regard what could be improved from the Weaknesses section:
- Why not include fully supervised baselines?
- Why not include error bars?
- What is the impact of the synthetic anomaly generation process in the results? Would they change outcomes if real spoofing doesn't follow the patterns seen in training?

---

### Official Review · Reviewer_dq6d · 2025-11-02

**Soundness:** 2
**Presentation:** 3
**Contribution:** 2
**Rating:** 2
**Confidence:** 5

**Summary:**

The paper studies manipulation detection using multilevel LOB data and proposes a two-stage framework, representation learning with  cascaded LOB encoding and supervised contrastive training, and an anomaly detector (OC-SVM and Isolation Forest). Experiments on LOBSTER data report gains over several representation backbones.

**Strengths:**

+ Manipulation detection is an interesting and practically important problem for financial markets, with clear societal impact.
+ Presents a representation-learning framework tailored to LOB data, aiming to leverage multilevel (depth >1) structure during training.

**Weaknesses:**

- Technical novelty is limited: the sequence model is essentially a Transformer, however the LOB’s rich information is not clearly captured the behavioral patterns in trade manipulation.
- The two-stage pipeline relies heavily on label quality, which raises concerns about robustness and practical deployability.
- Experimental design mainly contrasts different representation learners under the same anomaly detectors; this only shows representation quality indirectly. It remains unclear how the method fares against end-to-end manipulation detectors trained to directly predict manipulation from raw LOB (or weak labels).

**Questions:**

1. Can you provide an end-to-end baseline trained directly for manipulation labels?
2. How sensitive is performance to label noise or imperfect injection rules? It is suggested to evaluate detector robustness with label noise and domain shifts.

---

### Note · Authors · 2025-11-12

I have read and agree with the venue's withdrawal policy on behalf of myself and my co-authors.